# Asking Clarification Questions to Handle Ambiguity in Open-Domain QA

**Dongryeol Lee**[1*]        **Segwang Kim**[2*‡]        **Minwoo Lee**[1]

**Hwanhee Lee**[3]        **Joonsuk Park**[4,5,6]        **Sang-Woo Lee**[4,5,7]        **Kyomin Jung**[1†]

[1]Dept. of ECE, Seoul National University, [2]Samsung Electronics Mobile eXperience,
[3]Chung-Ang University, [4]NAVER AI Lab, [5]NAVER Cloud,
[6]University of Richmond, [7]KAIST AI

{drl123, ksk5693, minwoolee, kjung}@snu.ac.kr, hwanheelee@cau.ac.kr
park@joonsuk.org, sang.woo.lee@navercorp.com

## Abstract

Ambiguous questions persist in open-domain question answering, because formulating a precise question with a unique answer is often challenging. Previous works have tackled this issue by generating and answering disambiguated questions for all possible interpretations of the ambiguous question. Instead, we propose to ask a clarification question, where the user's response will help identify the interpretation that best aligns with the user's intention. We first present CAMBIGNQ, a dataset consisting of 5,653 ambiguous questions, each with relevant passages, possible answers, and a clarification question. The clarification questions were efficiently created by generating them using InstructGPT and manually revising them as necessary. We then define a pipeline of three tasks—(1) ambiguity detection, (2) clarification question generation, and (3) clarification-based QA. In the process, we adopt or design appropriate evaluation metrics to facilitate sound research. Lastly, we achieve F1 of 61.3, 25.1, and 40.5 on the three tasks, demonstrating the need for further improvements while providing competitive baselines for future work.

## 1 Introduction

In open-domain question answering (ODQA), questions can often be interpreted in several ways, each with a distinct answer (Min et al., 2020; Zhang and Choi, 2021). For example, consider the question at the top of Figure 1. Though the question seems unambiguous, "*young Tom Riddle*" can mean "*young version in series 2*", "*child version in series 6*", or "*teenager version in series 6*". Such ambiguity needs to be resolved to correctly find the answer sought by the user.

Previous studies propose to handle ambiguous questions (AQs) by generating a disambiguated

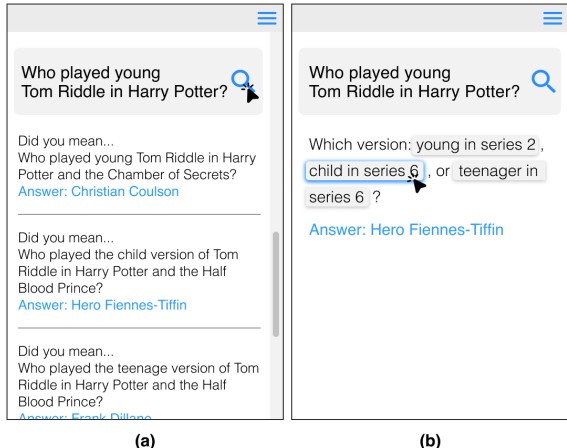

Figure 1: Two possible approaches to handling ambiguous questions (AQs) in open-domain question answering (ODQA): (a) presenting disambiguated questions (DQs) with answers (following Min et al. (2020)), and (b) asking a clarification question (CQ) and displaying an answer based on the user's response to the CQ (ours).

question (DQ; disambiguated variation of the given AQ) for each possible interpretation (Min et al., 2020; Gao et al., 2021; Stelmakh et al., 2022). While such DQ-based approaches are an important step toward resolving ambiguities in ODQA, imagine how it would be deployed in real life; without knowing the user's intention, the QA system would have to list all possible answers to the user, as shown in Figure 1(a). This is not suitable in most real-world scenarios where QA systems communicate with their users through speech or small-screen devices (Zamani et al., 2020b; Croft, 2019; Culpepper et al., 2018).

Instead, we propose to prompt the user with a clarification question (CQ), as shown in Figure 1(b). More specifically, given an AQ, the goal is to ask a CQ consisting of the possible interpretations as *options* (e.g. "teenager in series 6") along with a *category* summarizing the options (e.g. "version"). Ideally, the user's response to the CQ would help identify the interpretation that best aligns with the

---

*  Equal contribution.
†  Corresponding authors.
‡  Work done while he was in Seoul National University.

user's intention, and the corresponding answer can be presented to the user. This CQ-based approach is not only applicable in the aforementioned real-world scenarios, but also shown to be preferred by users according to our preference test. This is also consistent with the finding that asking CQs can improve user experience with "limited bandwidth" interfaces (Zamani et al., 2020a).

To support research on CQ-based approaches to handle AQs in ODQA, we present Clarifying Ambiguous Natural Questions (CAMBIGNQ). CAMBIGNQ is a dataset consisting of 5,653 AQs from AMBIGNQ (Min et al., 2020), each accompanied by relevant passages, possible answers, and a newly created CQ. The CQs were first generated using InstructGPT (Ouyang et al., 2022) through in-context few-shot learning, then manually vetted and edited as necessary by human editors. Such human-machine collaboration for constructing corpora has been shown to significantly reduce the time and the cost from fully manual approaches (Wang et al., 2021; Wu et al., 2021).

We also define a pipeline of three tasks to handle AQs in ODQA—(1) ambiguity detection, (2) clarification question generation, and (3) clarification-based QA. In the process, we adopt or design appropriate evaluation metrics to facilitate sound research. The experiments show that though they were shown to be helpful for generating DQs, predicted answers for AQ do not help improve the CQ-based approach overall. Lastly, we achieve F1 of 61.3, 25.1, and 40.5 on the three tasks, demonstrating the need for further improvements while providing competitive baselines for future work.[1]

Our main contributions are threefold:

- We propose to use CQs as a practical means to handle AQs in ODQA. Consistent with the findings by Zamani et al. (2020a), our human preference test shows that *the use of CQ is preferred over that of DQs* (Section 5).

- We present CAMBIGNQ, a dataset to support CQ-based handling of AQs in ODQA. It was built efficiently by leveraging a well-curated resource, AMBIGNQ, as well as the power of InstructGPT and human editors (Section 4).

- We define a pipeline of tasks and appropriate evaluation metrics for handling AQs in ODQA (Section 3). The experiments show

that though they were shown to be helpful for generating DQs, *predicted answers for AQ do not help improve the CQ-based approach overall* (Section 6).

## 2 Related Work

**Clarification Question Datasets** To resolve question ambiguity, CQ datasets have been released in various domains. In the information-seeking domain, CQ datasets for conversation (Aliannejadi et al., 2019, 2020; Guo et al., 2021; Wu et al., 2022) or web search (Zamani et al., 2020b) have been collected from crowdsourcing or real users' follow-up search queries. In the question-answering domain, datasets that focus on specific topics (Rao and Daumé III, 2018; Braslavski et al., 2017; Saeidi et al., 2018) or knowledge-base (Xu et al., 2019) has been proposed. To the best of our knowledge, we are the first to release a CQ dataset for ODQA.[2]

**Dataset Construction Leveraging LLMs** Manually constructing datasets from scratch is laborious and costly, which can be prohibitive depending on the nature of the dataset. Also, access to real users' data is strictly limited to a certain community. To mitigate these issues, approaches leveraging LLMs to construct datasets have recently been used in various domains such as dialogue (Bae et al., 2022), domain-adaptation (Dai et al., 2022), and in general (Ding et al., 2022). However, such an approach has not been used to construct CQ datasets, except for ours. We used InstructGPT (Ouyang et al., 2022) to generate CQs through in-context few-shot learning, and then manually vetted and edited them as necessary to construct our dataset.

**Clarification Question Evaluation** There are several options for evaluating the quality of CQs, First is leveraging widely-used automatic text evaluation metrics, such as BLEU (Papineni et al., 2002) or ROUGE (Lin, 2004). However, due to the poor correlations between such scores and human evaluation, Zamani et al. (2020b) strongly discourages the use of such metrics for evaluation. Second is human evaluation. While it typically provides a reliable estimate of how people would think of the given CQs, it can be time-consuming

---

[1]The data and code will be available at `https://github.com/DongryeolLee96/AskCQ`

[2]Xu et al. (2019) presents a CQ dataset for KBQA that is also open-domain, but the setting is much more restricted than ODQA in that the goal is to find the appropriate knowledge-base entry between exactly two entries about the same entity. Once an entry is determined, answering the question involves a simple table lookup, rather than a full-on QA.

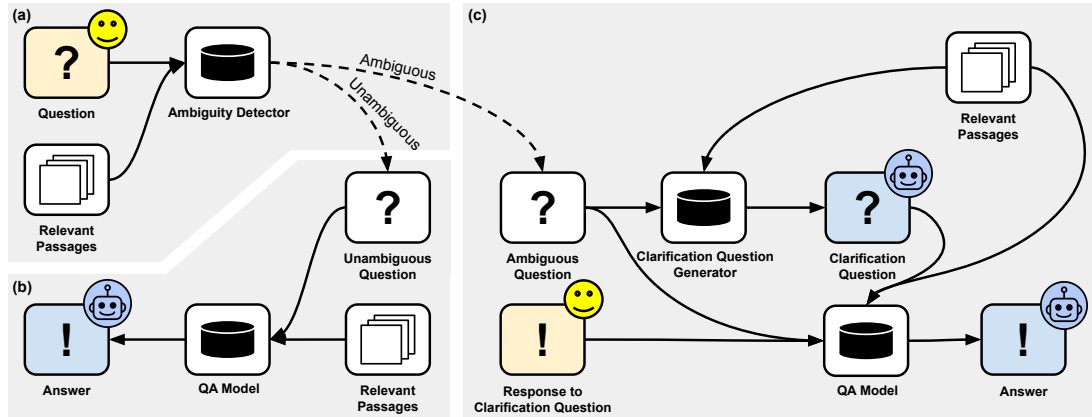

Figure 2: Overview of our proposed approach to ODQA. Given a question, it is first checked for ambiguity ((a) ambiguity detection). If it is not ambiguous, it is processed in a normal QA setup ((b) QA; outside the scope of this work). Otherwise, an extra step of eliciting a response to a clarification question precedes QA ((c) CQ generation + clarification-based QA). Yellow blocks represent the user input, and blue blocks, the system output.

and costly. As a third option, evaluation methods using external neural models have recently been introduced (Rei et al., 2020; Mehri and Eskenazi, 2020; Lee et al., 2021). This approach improves on the first, without the burden of the second. In this work, we design evaluation methods that suit our tasks, leveraging external neural models to provide a more comprehensive and accurate assessment.

## 3 Task Overview

We propose to handle AQs in ODQA by asking CQs as shown in Figure 2. There are three tasks: (1) *ambiguity detection*, (2) *clarification question generation*, and (3) *clarification-based QA*.

### 3.1 Task 1: Ambiguity Detection

**Task** Given a question and relevant passages, the goal is to determine whether the question is ambiguous or not, as shown in Figure 2(a). A question is considered ambiguous if it can be interpreted in multiple ways, each of which is associated with a unique answer, resulting in multiple possible answers. A question is unambiguous if it can only be interpreted in one way, thus one possible answer.

**Evaluation** For this binary classification task, we use standard metrics, such as accuracy, recall, precision, and F1.

### 3.2 Task 2: Clarification Question Generation

**Task** Given an AQ and relevant passages, the goal is to generate a CQ such that a valid response to the CQ is associated with exactly one of the multiple answers to AQ.

A CQ is typically formatted as follows:

"*Which* [category]*:* [option_1], [option_2], ..., or [option_n]?*"

Here, [category] is a category to which all options belong, such as "version" in Figure 1. If the options can not be grouped into a single category, "one" is used as a placeholder for the category. Also, where suitable, additional words like prepositions can precede "Which," e.g. "In which context". There should be an [option_j] for each possible interpretation of the AQ. Also, only the options are considered valid responses to the given CQ.

**Evaluation** We evaluate the quality of the generated CQ in two levels of granularity. First, we compare the generated CQ against the reference CQ using the standard BLEU-4 metric and BERTSCORE (Zhang et al., 2019).

Second, we evaluate the category and the options separately for a more fine-grained evaluation. For the category, exact match (EM) and BLEU-1 are computed since the category is typically very short. For the options, we adopt and adjust the partial match evaluation metric by Li et al. (2022), whose goal is to measure the similarity between a predicted set and a reference set of strings. Since the exact alignment of the strings between the sets is unknown, it measures the similarity—based on the longest common substring (LCS)—between all pairs across the sets and keeps the highest score. Here, multiple strings from the predicted set can be aligned with the same string in the reference set. In this work, we impose a constraint that limits the alignment of a reference option to at most one predicted option, since each option should represent a unique interpretation of the AQ. Thus, we find the

optimal assignment that maximizes the similarity score using the Hungarian algorithm (Kuhn, 1955) and compute precision, recall, and F1 as follows:

$$\max_i^{prec} = \sum_{p \in P_i} sim(p, f_i(p)), \qquad (1)$$

$$\max_i^{rec} = \sum_{r \in R_i} sim(r, f_i^{-1}(r)), \qquad (2)$$

$$\text{prec} = \frac{\sum_i \max_i^{prec}}{\sum_i |P_i|}, \text{rec} = \frac{\sum_i \max_i^{rec}}{\sum_i |R_i|}, \qquad (3)$$

where $P_i$ and $R_i$ is the set of predicted and reference options for $i$-th sample, $sim(\cdot)$ is the LCS-based similarity measure, and $f_i : P_i \rightarrow R_i$ is the optimal one-to-one mapping computed from the Hungarian algorithm. F1 is a harmonic mean of precision and recall, as usual.

Please refer to Appendix A.1 for more details.

### 3.3 Task 3: Clarification-based QA

**Task** Given an AQ, relevant passages, and a CQ, the goal is to generate a unique answer for every valid answer to the CQ—i.e., an option—which is associated with an interpretation of the AQ.

Each answer is generated by calling a QA model on an *AQ revised by CQ*, which is the concatenation of AQ, category, and single option:

"*AQ, Which [category]: [option_j]*".

Note, because each *AQ revised by CQ* is a unique interpretation of the AQ with a distinct answer, the relevant passages first need to be reranked before generating an answer.[3]

**Evaluation** The procedure is similar to that of option evaluation for CQ generation, in that it uses the partial match method with the Hungarian algorithm to determine the optimal alignment between predicted and reference answers.

The only difference is that $\max_i^{prec}$ and $\max_i^{rec}$ for each aligned pair of predicted and reference answers are computed differently, because in QA, the correct answer may be expressed in multiple ways, e.g., "Michael Jordan", "MJ", and "Jordan". Thus, a predicted answer is compared with all variations of the same answer, and the max score is used for that pair. Then, precision, recall, and F1 are calculated as before, with the newly computed $\max_i^{prec}$ and $\max_i^{rec}$.

Please refer to Appendix A.2 for more details.

---

[3] We utilized cross encoder MiniLMv2 model (Wang et al., 2020) fine-tuned on MSMARCO. https://huggingface.co/cross-encoder/ms-marco-MiniLM-L-12-v2

| Split | CQ | | Category | Options | |
|---|---|---|---|---|---|
| | # | Len. | Len. | Avg. # | Len. |
| Train | 4,699 | 13.6 | 2.8 | 2.9 | 3.3 |
| Validation | 461 | 15.9 | 2.5 | 3.3 | 3.8 |
| Test | 493 | 17.8 | 2.8 | 3.4 | 4.1 |

Table 1: Statistics of CAMBIGNQ. Each clarification question (CQ) consists of one category and multiple options. The length is reported in the number of words.

## 4 The CAMBIGNQ Dataset

### 4.1 Dataset Construction

We present Clarifying Ambiguous Natural Questions (CAMBIGNQ), a new dataset consisting of 5,653 AQ, each with relevant passages, possible answers, and a CQ. CAMBIGNQ was constructed from the AQs in AMBIGNQ (Min et al., 2020), which provides each AQ with relevant passages, as well as the DQ and answer pairs reflecting the possible interpretations and respective answers of the AQ. To build CAMBIGNQ, we replaced each set of DQs with a CQ. In other words, the CQ is an integrated version of the set of DQs. Representing each DQ as a single phrase option can be cumbersome to do manually. Thus, we collect high-quality of CQs by leveraging InstructGPT, using a two-step framework: *Generation via InstructGPT* and *Manual Inspection and Revision*.

**Generation via InstructGPT** To take advantage of the few-shot learning capability of InstructGPT, we first manually annotate a small number of CQs for AQs. These edited CQs are then used as "few-shot" examples along with brief instructions and both the AQ and corresponding DQs. We sampled *six* examples considering the diversity of *category* and *number of options*. The final prompt is in the form of a concatenation of the six examples, instructions, target AQ, and target DQs in the following form:

"*instruction, AQ_1, DQs_1, CQ_1, ..., instruction, AQ_6, DQs_6, CQ_6, instruction, AQ_target, DQs_target*"

**Manual Inspection and Revision** The recruited annotators were asked to read instructions and revise 25 CQs accordingly as a qualification test. Then the editors who passed the qualification test were asked to examine, and revise as necessary, the CQs generated by InstructGPT.

They were asked to follow the following protocol to ensure the quality of the final CQs: First, check whether the AQ had at least two distinct inter-

| Target | Question | Example |
|---|---|---|
| Category | AQ | Who is Catch Me If You Can based on? |
| Only | GPT CQ | Which one: the 2002 film, the book, or the musical? |
| (12.9%) | Edited CQ | Which version: the 2002 film, the book, or the musical? |
| Options | AQ | When did the £20 note come out? |
| Only | GPT CQ | Which series: F, or E? |
| (19.7%) | Edited CQ | Which series: F, E variant, or E? |
| Category | AQ | Who plays Will on The Bold and Beautiful? |
| & Options | GPT CQ | Which time period: first, replacement, or 2013? |
| (31.4%) | Edited CQ | Which one: first actor, actor that replaces the wardens, or actor that began playing in 2013? |
| Whole | AQ | Who is the all-time passing leader in the NFL? |
| Question | GPT CQ | Does the leader include regular season stats, or stats from the playoffs as well? |
| (7.8%) | Edited CQ | In which context: in the regular seasons, or including the playoffs as well? |
| None | AQ | Who is the current chairman of African Union commission? |
| (26.7%) | GPT CQ | Which chairman: 4th, 3rd, or 2nd? |
| | Edited CQ | Which chairman: 4th, 3rd, or 2nd? |

Table 2: Examples of manual revisions made to clarification questions (CQs) generated by InstructGPT for ambiguous questions (AQs). The human editors were provided with disambiguated questions (DQs) for reference. Red and blue words represent the words before and after revision, respectively. The remaining 1.5% was marked as "unambiguous" by the editors, meaning only one interpretation, and thus one answer, exists for the given question. These were excluded from our dataset.

pretations and corresponding DQs; Second, check whether the CQ generated by InstructGPT is in the correct format (See Section 3.2); Third, check whether each option accurately represents its corresponding DQ and the category is a correct term describing the set of options. The editors had three actions to choose from—they could either: mark the AQ as not ambiguous, i.e. there is only a single interpretation and answer (occurred in 1.5% of cases), revise the CQ (occurred in 71.8% of cases), or leave the CQ as is (occurred in 26.7% of cases). (See Table 2 for example revisions.) The high revision rate suggests that the few-shot generation with InstructGPT is far from perfect, and manual editing was necessary.

For inter-annotator agreement, we use the validation set which was annotated by two annotators, following Min et al. (2020). The kappa coefficient (Cohen, 1960) is 0.623, which can be considered a "substantial agreement." (McHugh, 2012).

### 4.2 Dataset Analysis

The entire dataset consists of 5,653 data points, as shown in Table 1. The training set was sourced from that of AMBIGNQ, while the validation and test sets were randomly split from the development set of AMBIGNQ. Each AQ in the dataset has over three interpretations on average, which in turn means that each CQ has over three options on average. The average length of the CQs varies from one split to another, with a general trend of longer CQs having more options.

The first column in Table 2 shows the statistics

on which components of the CQs generated by InstructGPT were revised by human editors. Of the entire dataset, about 8% were due to the invalid format of the CQs. This means that although InstructGPT was provided with six example CQs in the prompt, it is not always enough to enforce the format. Additionally, one common type of revision made to the category was converted to or from "one", meaning InstructGPT often tried to group ungroupable options or chose not to group options that can be grouped into a single category. A common revision made to the options was to split what InstructGPT generated as a single option. Errors like this also lead to a mismatch between the number of DQs, or interpretations, and that of the options. Overall, there seems to be room for further prompt engineering to minimize errors, but we believe manual revisions are a necessary component for constructing high-quality datasets as of yet. Please refer to Appendix B for more details.

## 5 Experiment 1: CQ vs DQ Preference

We first conduct a human preference test to investigate the question: *Is our CQ-based approach preferred over a DQ-based one to handle AQs in ODQA?* This is to check if it is worthwhile to pursue the CQ-based approach.

**Setup** We randomly sampled 100 AQs from the development set. Then, for each AQ, we asked three annotators to show their preferences for "CQ", "DQ", or "Equal", along with the rationale. That is, given an AQ, we ask people to compare "being pre-

| CQ | Split | DQ |
|----|-------|-----|
| 0.59 | 0.08 | 0.33 |

Figure 3: Percentage of questions where the majority of people preferred "CQ", "DQ" and "Split", respectively. "Split" denotes that there was no majority response.

sented with answers to all possible interpretations (DQs) of the AQ" vs "first answering a CQ and then being presented with an answer fitting their intention." We then report the majority preference for each of the questions. Please refer to Appendix C for more details.

**Results and Analysis** Figure 3 demonstrates that answering AQs using CQ is preferred over DQ. The prominent reasons stated by annotators for favoring "CQ" are its ease of use, conciseness, interactivity, and ability to provide clear guidance. Conversely, annotators who preferred "DQ" mentioned its advantage as being more specific and clearer in addressing the given question.

Note, CQ was unanimously preferred 23 times, and DQ, 5 times. Also, unanimity in favor of CQ was observed across AQs regardless of the number of interpretations—or options—whereas unanimity in favor of DQ only occurred for AQs with up to three interpretations. In other words, CQ can be preferred regardless of the number of interpretations, while DQ is not preferred when many interpretations are possible. This is intuitive given that more interpretations result in more text for people to process for the DQ-based response to AQ.

## 6 Experiment 2: Handling AQ with CQ

Given that the CQs are preferred over DQs, we now study the question: *Do predicted answers for AQ help improved the end-to-end performance of the CQ-based approach?* Since predicted answers for AQ have been shown to be helpful for previous DQ-based approaches (Min et al., 2020; Gao et al., 2021), we want to verify if they are also helpful for the CQ-based approach.[4] For this, we experiment with two settings:

1. *Predicted Answers for AQ:* running a QA model on the AQ and incorporating the predicted answers as input to the subsequent tasks

---
[4]Note, while both CQ-based and DQ-based approaches seek to solve the issue of AQs in ODQA, the performances of the models are not directly comparable. This is because the problems are formulated differently—the former generates a CQ and an answer aligned with the user's intention, whereas the latter generates DQs and answers for them.

| Input in addition to AQ | Acc. | Pre. | Rec. | F1 |
|---|---|---|---|---|
| No Answers for AQ | **63.9** | **61.9** | **60.7** | **61.3** |
| Predicted Answers for AQ | 56.5 | 59.7 | 24.1 | 34.3 |

Table 3: Evaluation results for the Ambiguity Detection task. The *No Answers* case uses BERT-BASE-CASED to determine whether a given question is ambiguous or not. The *Predicted Answers* case makes use of answers predicted by SPANSEQGEN and classifies the question as unambiguous only if exactly one answer is predicted.

2. *No Answers for AQ:* not predicting answers to the AQ, and thus not using them in the subsequent tasks

In the remainder of the section, we present the experimental setup and results for each task. Please refer to Appendix D for more details.

### 6.1 Task 1: Ambiguity Detection

**Setup** Since our dataset consists only of AQs, i.e., questions with multiple interpretations and answers, we combine it with unambiguous questions, i.e., questions with a single interpretation and answer, from AMBIGNQ for this task.

For *No Answers for AQ* case, we use the BERT$_{BASE}$ model (Devlin et al., 2018) with a simple fully connected layer on top for the binary classification task. The model is trained on the combined dataset for 96 epochs. The model also takes in a prompt of the form "*question [SEP] relevant_passages*" as input and outputs "Ambiguous" or "Unambiguous".

For *Predicted Answers for AQ* case, we use BART-based model called SPANSEQGEN, the best-performing model for predicting answers for AQ by Min et al. (2020), and finetuned it on the AMBIGNQ dataset. This model takes in a prompt of the form "*question [SEP] relevant_passages*" as input and predicts all plausible answers. We classify a question as "Ambiguous" if the model outputs more than one plausible answer and "Unambiguous," otherwise.

**Results and Analysis** Table 3 summarizes the result of ambiguity detection of two models BERT$_{BASE}$ (*No Answers*) and SPANSEQGEN (*Predicted Answers*). SPANSEQGEN exhibits a similar precision as BERT$_{BASE}$ (59.7 vs 61.9) but a significantly lower recall (24.1 vs 60.7). This is because most questions are classified as "Unambiguous." since the average number of answers generated by SPANSEQGEN is 1.24. Consequently, this results in a much higher precision when compared to the

| Input in addition to AQ and RPs | CQ | | Category | | Options | | | |
|---|---|---|---|---|---|---|---|---|
| | BLEU-4 | BERTSCORE | EM | BLEU-1 | Pre. | Rec. | F1 | Avg. # |
| No Answers for AQ | **7.9** | **88.9** | 20.2 | **47.3** | **37.4** | 18.2 | 24.5 | 2.0 |
| Predicted Answers for AQ | **7.9** | **88.9** | **22.8** | 44.0 | 36.9 | **19.0** | **25.1** | 2.0 |
| Ground Truth Answers for AQ | 15.4 | 89.6 | 25.2 | 46.9 | 34.3 | 34.4 | 34.3 | 3.7 |

Table 4: Evaluation results for the Clarification Question (CQ) Generation task, where generated CQs are compared against the reference CQs. Each CQ was generated from an ambiguous question (AQ), relevant passages (RPs), and either *No Answers*, *Predicted Answers*, or *Ground Truth Answers* for the AQ. The *ground truth answers* case represents an ideal scenario in which the QA system perfectly identifies all possible answers for the AQ.

| CQ used to clarify the AQ | NQ-pretrained BART | | | | CQ-finetuned BART | | | |
|---|---|---|---|---|---|---|---|---|
| | Pre. | Rec. | F1 | # Ans. | Pre. | Rec. | F1 | # Ans. |
| CQ generated with No Answers for AQ | 47.9 | 25.2 | 33.0 | 1.5 | 54.4 | 31.1 | 39.6 | 1.6 |
| CQ generated with Predicted Answers for AQ | **49.6** | **26.2** | **34.3** | 1.5 | **55.4** | **32.0** | **40.5** | 1.6 |
| CQ generated with Ground Truth Answers for AQ | 39.7 | 37.5 | 38.6 | 2.0 | 47.5 | 49.5 | 48.5 | 2.5 |
| Ground Truth CQ | 47.5 | 39.8 | 43.3 | 2.0 | 58.0 | 53.8 | 55.8 | 2.5 |

Table 5: Evaluation results for the Clarification-based QA task. Answers found by a QA model for the AQs clarified with CQs are compared against the ground truth answers for the AQs. Three variations of model-generated CQs, derived from the CQ Generation task, are used to clarify the AQs. The *Ground Truth CQ* case is an ideal scenario in which Ground Truth CQs are used to clarify the AQs. The # Ans. is the average number of unique answers predicted for each AQ.

recall for the same case. This result indicates that classifying AQs by predicting all plausible answers is a challenging task for the Seq2Seq model.

## 6.2 Task 2: Clarification Question Generation

**Setup** For this task, we only use ground truth AQs to isolate the task from ambiguity detection. Please refer to Section 6.4 for experiments in which errors do propagate from one task to the next.

For *No Answers for AQ* , we first train a $BART_{large}$ model for 18 epochs, that takes "*AQ [SEP] relevant_passages*" as input and generates CQ as output. During inference, this model was used with a prompt of the form "*AQ [SEP] relevant_passages*".

For *Predicted Answers for AQ*, we train another $BART_{large}$ model for 41 epochs, that takes "*AQ [SEP] possible_answers [SEP] relevant_passages*" as input and generates a CQ as output. During inference, the model takes input with *possible_answers* as answers predicted by SPANSEQGEN.

We also consider an additional setting, the *Ground Truth Answers for AQ* case. This case is an ideal version of the *Predicted Answers for AQ* case, where the ground truth answers are used as *possible_answers*. Hence, this case allows us to examine the effect of providing the correct set of plausible answers.

**Results and Analysis** The evaluation results of CQ generation with three variations of inputs are presented in Table 4. The results indicate that in

the two realistic scenarios (*No Answers for AQ*, *Predicted Answers for AQ*), the quality of the generated CQs does not vary significantly in terms of the CQs themselves, the category, or the options. This suggests that incorporating plausible answers as input to the CQ generation process does not improve the quality of the generated CQs in realistic scenarios. However, when ground-truth answers are provided as input to the CQ generation process (*Ground Truth Answers for AQ*), a significant improvement in the quality of the generated CQs is observed, but the quality is seemingly insufficient with a large room for improvement.

In some cases, predicted CQs that are semantically correct were counted as incorrect. For example, the model generated the CQ "Which chairman: 2017 or 2012?" for example 5 in Table 2. Although deemed incorrect, a manual examination of relevant passages revealed the 4th and 3rd chairmen took office in 2017 and 2012, respectively. This illustrates the challenge of directly comparing a predicted CQ and the respective reference CQ. Thus, the absolute score in Table 4 may not be as meaningful as the relative scores. Also, evaluating CQs in a downstream task may be necessary to better assess the qualities of the CQs, which we do in the clarification-based QA task.

## 6.3 Task 3: Clarification-based QA

**Setup** We use NQ-pretrained $BART_{large}$ for the reader model which was trained on Natural Ques-

tions (NQ) dataset ([Kwiatkowski et al., 2019a](#)). The model takes in an *AQ clarified by CQ*—which is the concatenation of AQ, category, and option—and reranked relevant passages as input and predicts an answer for *AQ clarified by CQ*. (See Section 3.3 for more details on *AQ clarified by CQ*)

In addition to the NQ-pretrained model, we also finetuned the NQ-pretrained reader model (CQ-finetuned BART) on our proposed dataset for 8 epochs. During finetuning, the model also takes in an *AQ clarified by CQ* as input. The target label is the corresponding answer for each option.

During the inference, we employed three variations of model-generated CQs, derived from Section 6.2 CQ Generation task. Moreover, we consider an ideal scenario wherein the Ground Truth CQ is available and used to clarify the AQ.

**Results and Analysis** The evaluation results of clarification-based QA using four variations of input and different reader models are presented in Table 5. Two ideal settings (*CQ generated with Ground Truth Answers for AQ* and *Ground Truth CQ*) exhibit lower precision scores. On the other hand, they outperform the other two variations (*CQ generated with No Answers for AQ* and *CQ generated with Predicted Answers for AQ*) in terms of recall, resulting in higher F1 scores, as well. One reason for this is that the *CQs generated by Ground Truth Answers for AQ* and *Ground Truth CQs* contain more options (1.5 more on average) which leads to predicting more answers than the other two variations, resulting in higher recall and lower precision scores.

The average numbers of options in Table 4 and those of unique answers in Table 5 indicate that both NQ-pretrained BART and CQ-finetuned BART struggle to generate distinct answers for distinct options. For instance, in the *CQ generated with Ground Truth Answers for AQ* case, where the average number of options for CQs is 3.7, only 2.5 distinct answers were generated for each AQ. In other words, both models tend to produce the same answer for the given AQ even if the specified options are different. This phenomenon, referred to as the "collapse" of the models has also been reported in previous studies such as ([Zhang and Choi, 2021](#)). It suggests that deep learning models can be insensitive to subtle differences in the input—when different options are chosen for the same AQ, the input would be identical except for the option.

| Ambig. Detect. | CQ Gen. | Pre. | Rec. | F1 |
|---|---|---|---|---|
| No Answers | No Answers | **43.2** | **19.9** | **27.3** |
| | Pred Answers | 42.8 | 19.6 | 26.9 |
| Pred Answers | No Answers | 22.5 | 8.3 | 12.1 |
| | Pred Answers | 24.7 | 9.0 | 13.1 |

Table 6: End-to-end Evaluation Results: The performances are measured at the end of the pipeline, i.e., clarification-based QA.

## 6.4 End-to-End

**Setup** We now conduct experiments to check RQ2—whether predicted answers for AQ help improve the CQ-based approach to handle AQ end-to-end. We consider four combinations of setting for ambiguity detection and CQ generation:

1. *Pred Answers–Pred Answers*: running a QA model on the AQ and incorporating the predicted answers in both tasks

2. *No answers–No Answers:* not running the QA model on AQ

3. *Pred Answers–No Answers:* running the QA model on the AQ but using the predicted answers as input for ambiguity detection only

4. *No Answers–Pred Answers:* running the QA model on the AQ but using the predicted answers as input for CQ generation only

The end-to-end performances are measured at the end of the pipeline, i.e., clarification-based QA.

**Results and Analysis** As shown in Table 6, the use of the BERT model (*No Answers*) for ambiguity detection and prompting without answers (*No Answers*) in the input for CQ generation yields the highest F1 score of 27.3. However, the combination of using the BERT model (*No Answers*) for ambiguity detection and utilizing predicted answers by the SPANSEQGEN model (*Pred Answers*) in the input for CQ generation resulted in an F1 score 0.4 lower than the best combination. Note, the *No Answers–Pred Answers* setting is not only (slightly) worse than the best approach, but is also inefficient as it requires running both BERT and SPANSEQGEN models during inference.

*No Answers–No Answers* and *Pred Answers–Pred Answers* are the only settings in which only a single model is used for ambiguity detection and generating input for CQ generation. Among these, the quality of the generated CQs varies significantly. More specifically, the results show that in the *Pred Answers–Pred Answers* scenario, the poor performance of the ambiguity detection stage propagates

to the remainder of the pipeline. This suggests that incorporating plausible answers as input to the CQ generation process prior to generating the CQs is not a desirable approach in the CQ framework. Finally, the end-to-end performance of all four cases still has a large room for improvement, showing the challenging nature of CQ-based approach to handling AQs in ODQA, as well as the need for resources like CAMBIGNQ.

## 7 Conclusion

We proposed a CQ-based approach to handle AQs in ODQA. Along with presenting a new dataset, we defined a pipeline of tasks, designing appropriate evaluation metrics for it. Experiments show the promising, yet challenging nature of the tasks. We hope that our dataset will serve as a valuable resource for research in this area and support the development of more user-friendly QA systems.

## Limitations

As shown in our results, both clarification question generation and clarification-based question answering evaluations can still underestimate the performance of the generated clarification questions due to various factors. One reason is that the reference clarification questions are one of many possible answers, not the only correct answer. Another reason is that the intrinsic evaluation, which depends on the overlap between the texts, may not properly handle semantically correct predictions. Additionally, the extrinsic QA model for clarification-based question answering may fail to perform reasoning. These limitations highlight the need for further research in the field to improve evaluation methods for clarification question generation tasks.

## Ethics Statement

Our proposed datasets will not pose any ethical problems as they have been constructed from the publicly available AMBIGNQ (Min et al., 2020) dataset, which itself is derived from the Natural Questions dataset (Kwiatkowski et al., 2019b).

Additionally, the use of the InstructGPT model for the generation of data was done by utilizing the official website of OpenAI[5]. All models used in the experiments are from the publicly available website or Github. While there is a possibility of bias or toxicity in the generated text, such issues are

---

[5] https://openai.com/

addressed through our human validation process. Furthermore, the data annotators were fairly compensated for their work, and the details of payment can be found in Appendix B.

## Acknowledgements

K. Jung is with ASRI, Seoul National University, Korea. This work has been financially supported by SNU-NAVER Hyperscale AI Center. This work was partly supported by Institute of Information & communications Technology Planning & Evaluation (IITP) grant funded by the Korea government(MSIT) [NO.2021-0-01343, Artificial Intelligence Graduate School Program (Seoul National University)]

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

# A Details of a partial match with the Hungarian algorithm

## A.1 Alignment for Clarification Question Generation task

The similarity function $sim(x, y)$ is defined as follows:

$$sim(x, y) = len(LCS(x, y))/len(x) \quad (4)$$

In Equation 1 and 2, similarity scores between prediction and reference are calculated by dividing the length of the longest common subsequence by the length of the predicted option and reference option, respectively.

To match each predicted option to a corresponding reference option, the Hungarian algorithm (Kuhn, 1955)[6] is applied, and optimal correspondence mapping function $f_i$ for $i$-th option is obtained.

## A.2 Alignment for Clarification-based Question Answering

In the evaluation for CBQA, the reference answers are not a single string but rather a list of strings that may represent a single answer. In this sense, the $\max_i^{prec}$ and $\max_i^{rec}$ for $i$-th example is calculated differently as follows:

$$\max_i^{prec} = \sum_{p \in P_i} \max_j sim(p, f_i(p)_j), \quad (5)$$

where, $f_i(p)_j$ is the single representation (e.g. "MJ" for the example from Section 3.3) of the reference answer set and prediction p is aligned to a single reference answer set consisting of total $J$ representations.

$$\max_i^{rec} = \sum_{r \in R_i} \max_j sim(r_j, f_i^{-1}(r)), \quad (6)$$

where, r is not the single string but a list of J strings that may represent a single answer. (e.g. ["MJ", "Michael Jefferey Jordan", "Jordan"] for the example from Section 3.3) and all representations in the list r are aligned to the same prediction by $f_i^{-1}(r)$.

---

[6]The Hungarian algorithm, used for assigning tasks to workers in a one-to-one manner with the objective of minimizing the cost, is adapted in our study to maximize the cost by altering the setting.

## B  Details of data collection and dataset

### B.1  Details of data generation by LLM

We use OpenAI (text-davinci-003) API model for the generation. For the hyperparameters of Instruct-GPT, we limited the *maximum length* of the output token to 100 and used 0.7 for *temperature*. We set *top_p* and *n* to 1. As mentioned in Section 4, the prompt is in the form of a concatenation of the six examples, instructions, target AQ, and target DQs. Each example used for the prompt is described in Table 9.

### B.2  Details of Manual Editing

Ten undergraduate students fluent in English were recruited through the university's online community, and seven of them successfully passed the qualification test. The recruited annotators were provided with a detailed description of task definitions, instructions, and examples as shown in Figure 4, 5. During the recruitment process, all applicants were informed that their annotations would be utilized for academic purposes and would be included in published papers. This information was explicitly stated in the recruitment announcement and instructions to ensure transparency. The annotators were then asked to review 25 examples that had been previously annotated by co-authors and revise 25 CQs generated by the InstrcutGPT model.

Seven annotators who passed the qualification stages were then selected to participate in the manual editing stages. As shown in Figure 4, the annotators were provided with *Ambiguous Question* and *Disambiguated Questions* on the left side of the page. To assist the annotation process, we used a process to identify the longest common subsequence between the AQ and DQs, a *spaCy* constituency parser to identify different constituent parts of the DQs, and highlighted these parts. On the right side of the page, the InstructGPT-generated CQ was provided, and the annotators were given the option to revise, pass (no revision), or report (single interpretation and answer) the given CQs.

We used streamlit[7], an open-source app framework for creating web apps for data science and machine learning, to construct the interface. The InstructGPT-generated examples were divided into sets of 500 examples, and for quality control, we in-

---

[7] https://streamlit.io/

cluded 20 validation examples that were annotated by a co-author in each set. The annotators were notified of the existence of the validation examples and asked to re-annotate the samples if the correct percentage of the correctly annotated validation examples did not meet a pre-determined threshold.

For the payment of the annotators, the co-authors first conducted annotations for an hour to estimate the average number of annotations that could be completed within an hour. Based on this estimation, a rate of 0.15 dollars per example was established to ensure that the annotators would be paid at least 133% of the minimum wage.

### B.3  Details of Inter-Annotator Agreement

We conducted an evaluation of the output from two annotators and report the BLEU score and EM score. The BLEU score and EM score for the entire CQ are 65.8 and 39.3, respectively. When considering the category and options separately, the BLEU score for the category is 76.5 with an EM score of 56.8, while the BLEU score for options is 66.0 with an EM score of 63.9. All scores provided have been micro-averaged.

We conduct an analysis of instances in which there was disagreement between the two annotators. The primary cause of these disagreements can be attributed to variations in the specificity of categories, or the options provided. For example, in Figure 1, there were different opinions within the category of "version" and the first option "young in series 2". The other annotator suggested alternatives such as "version of Tom Riddle" for the category and "the young Tom Riddle in Harry Potter and the Chamber of Secrets" for the first option. These alternatives are accurate in capturing the intended meaning, but they differ in their surface form.

### B.4  Details of human editing

We provide a deeper analysis of human editing made on examples generated by InstructGPT, as shown in Table 2. Specifically, for instances where human annotators made partial revisions, focusing on either the category or the options alone (referred to "Category Only" and "Options Only" in Table 2), we compute the BLEU score. Interestingly, both the "Category Only" and "Options Only" cases exhibit BLEU scores of 37.0 and 53.6, respectively. Additionally, in cases where the model generated invalid forms (referred to "Whole Question" in Table 2), the BLEU scores between the model's

predictions and the human revisions yield 36.7. It is important to note that while BLEU scores may not capture semantic similarity, they do provide valuable insights into the disparity between human-labeled data and model-generated data.

## B.5 Details of dataset

Table 10 provides an overview of the most frequently used categories within each split of the dataset. It is evident that the top five categories consistently appear in all three sets, although their specific order may vary. This suggests that the dataset was well split into three parts. Moreover, Figure 6 illustrates the top 50 categories from the entire dataset, providing a broader perspective. Overall, we have 593 distinct categories, with 412 of these categories occurring only once. This observation aligns with the previously discussed Inter-Annotator Agreement in Section B.3, where variations in specificity among annotators contribute to the presence of unique categories. For instance, examples such as "Jurrassic world" vs "movie" or "Will Turner" vs "character" illustrate this inherent variability. This existence of multiple representations for single categories or options is inherent to our task, and it is considered a natural occurrence. The set of categories can be expanded as needed. Additionally, we provide the number of options in our dataset in Figure 7. Since our dataset is built upon the AMBIGNQ, the distribution of options is comparable to the previous study.

## C  Details of human preference test

### C.1  Details of test setup

We use Amazon Mechanical Turk[8] for the human preference test. To ensure the quality of responses, we restricted the workers whose nations are the US, CA, AU, NZ, UK and whose HIT minimum hits are over 5,000, and HIT rates are higher than 97%. Additionally, we enforced a requirement for annotators to provide at least one sentence explaining the reason for their choices. Instances where annotators failed to provide a reason, provided a reason consisting of few words or presented a reason irrelevant to our task were deemed as "rejected" cases. Annotators were informed of this rule and compensation for the MTurk workers was set at more than $10 per hour.

In order to investigate the potential correlation between the number of interpretations (i.e., the

number of DQs) and user preference, we proceeded by partitioning our development dataset into five distinct groups based on the number of interpretations. These groups were categorized as data with two, three, four, five, and more than five interpretations. Subsequently, a single example was randomly selected from each group, resulting in the formation of one batch comprising five instances. Each batch was then assigned to an annotator for annotation, with a total of 20 batches being processed in this manner. An example of annotation interfaces is shown in Figure 8.

### C.2  Detailed analysis on test result

Figure 9 provides the results obtained for varying numbers of interpretations. The percentages of annotators favoring "CQ" remain relatively stable across different numbers of interpretations.

## D  Training Details

**Training Detail of Ambiguity Detection**  The Ambiguity Detection task utilized a combined dataset consisting of 9,996, 977, and 977 instances in the train, validation, and test sets, respectively. For the model, the BERT-base-cased model[9] was finetuned with *batch_size* 16, *accumulation_step* 1, *learning rate 2e-5*, and *early_stop_patience* 1. We use released checkpoint for pretrained SPANSE-QGEN model[10]. We used one A5000 GPU for finetuning and it took approximately 4 hours. The training epochs were determined according to the validation performance based on accuracy.

**Training Detail of Clarification Question Generation**  The two BART_large[11][12] were finetuned on our CAMBIGNQ with the training/validation/test split as described in Table 1. Both models share the same hyperparameter during finetuning, which are *batch_size* 10, *accumulation_step* 2, *learning rate 1e-5*, and *early_stop_patience* 10. The training epochs were determined according to the validation performance based on the BLEU score of the whole CQ. We used one A6000 GPU for both finetuning and it took approximately 2 hours for *No Answers* case and 4 hours for *Predicted Answers* and *Ground Truth Answers* cases.

---

[8]https://www.mturk.com/

[9]https://huggingface.co/bert-base-cased

[10]https://nlp.cs.washington.edu/ambigqa/models/ambignq-bart-large-12-0.zip

[11]https://huggingface.co/facebook/bart-large

[12]the total length of input was truncated to 1,024 tokens due to the maximum input length of the model

| Reader Model | Pre. | Rec. | F1 | Acc. |
|---|---|---|---|---|
| CQ finetuned BART | **58.0** | 53.8 | **55.8** | 35.8 |
| InstructGPT | 7.4 | **60.0** | 13.1 | **43.2** |

Table 7: Evaluation results for the Clarification-based QA task employing two different reader models. Both cases utilize the truth CQs to clarify the AQ. The Acc. represents *accuracy* which evaluates whether the model's response includes any gold answer.

**Training Detail of Clarification-Based Question Answering**   For Clarification-Based Question Answering, the NQ-pretrained BART model[13] was finetuned with *batch_size* 10, *accumulation_step* 2, *learning rate 1e-5*, and *early_stop_patience* 10. We used one A6000 GPU for finetuning and it took approximately 2 hours.

# E   Inference Employing Large Language Models

Our primary experiments, which leveraged the BART-large models as our baselines, demonstrated suboptimal performance across different settings. To evaluate the efficacy of recent Large Language Models (LLMs) in our task, we designed additional experiments incorporating LLMs within our framework. In these supplementary experiments, we only consider the most ideal case from Section 6.3 where Ground Truth CQs are available.

**Setup**   We leveraged two distinct variations of InstructGPT (Ouyang et al., 2022), provided by OpenAI (namely, text-davinci-003, gpt-3.5-turbo), for our additional studies. Initially, we employed the text-davinci-003 model (InstructGPT) as the reader model in Section 6.3, replacing the previously used BART-large models. Subsequently, we reformulated our task as an interactive dialogue between the user and the QA models, comprising the following sequence: 1) User asking AQ, 2) Model offering CQs, 3) User selecting an option, and 4) Model generating corresponding answer for a given option. Within this conversational framework, we utilized the gpt-3.5-turbo model (ChatGPT) and conducted inference under two settings: zero-shot and four-shot.

We evaluate both models using the conventional metrics of precision, recall, and F1 score. Furthermore, due to the fact that both models generate responses at the sentence-level, quantifying the

| ChatGPT | Pre. | Rec. | F1 | Accuracy |
|---|---|---|---|---|
| Zero-shot | 8.0 | **64.5** | 14.3 | **50.8** |
| Four-shot | **11.3** | 64.0 | **19.2** | 49.9 |

Table 8: Evaluation results of conversational setting employing ChatGPT. ChatGPT receives the input framed as an interactive dialogue between the user and the model, outlined in the subsequent sequence: 1) User asking AQ, 2) Model offering ground truth CQs, 3) User selecting an option, and 4) Model generating the corresponding answer for a selected option. Zero-shot and Four-shot denote the number of examples presented to the model within the prompt.

number of unique answers is challenging. Following Liu et al. (2023) and Mallen et al. (2022), we adopt *accuracy*, which evaluates whether the prediction includes any gold answer.

**Results and Analysis**   The evaluation results of clarification-based QA, utilizing InstructGPT as the reader model, and our task's reformulation within an interactive dialogue framework with ChatGPT, are presented in Table 7 and Table 8, respectively. It is noteworthy that the low precision in both results from InstructGPT, ChatGPT in zero-shot configuration, and ChatGPT in four-shot configuration can be attributed to the model's tendency to generate answers at the sentence level. These responses average 27.3 words, 25.4 words, and 19.9 words respectively. In contrast, the gold answers are more concise, averaging 2.6 words, leading to the observed low precision scores.

Utilizing InstructGPT as a reader model showed improved performance compared to our baseline, which uses CQ fine-tuned BART as a reader model. Additionally, reformulating our task as an interactive dialogue and incorporating ChatGPT shows improved recall and accuracy. However, it is evident that there is substantial potential for further enhancement which underscores both the challenging nature of our tasks and the need for further research.

---

[13]https://nlp.cs.washington.edu/ambigqa/models/nq-bart-large-24-0.zip

| | |
|---|---|
| Instruction | Generate the clarifying question for an ambiguous question that gives options for corresponding disambiguated question. |
| Example_1 | ambiguous question: Why did the st louis cardinals move to arizona?
disambiguated question 1: what ability caused the st louis cardinals move to arizona?
disambiguated question 2: what physical issue caused the st louis cardinals move to arizona?
disambiguated question 3: what fan issue caused the st louis cardinals move to arizona?
clarifying question: Which type of reason: Ability, physical issue, or fan issue? |
| Example_2 | ambiguous question: Who is the current chairman of african union commission?
disambiguated question 1: who is the 4th chairman of african union commission?
disambiguated question 2: who is the 3rd chairman of african union commission?
disambiguated question 3: who is the 2nd chairman of african union commission?
clarifying question: Which chairman: 4th, 3rd, or 2nd? |
| Example_3 | ambiguous question: Who won the final hoh big brother 20?
disambiguated question 1: who won the final hoh in the american reality show big brother 20?
disambiguated question 2: who won the final vote in the british reality show celebrity big brother 20?
clarifying question: Which version: the american reality show, or the british reality show celebrity? |
| Example_4 | ambiguous question: How long do contestants get to answer on jeopardy?
disambiguated question 1: how long do contestants get to answer a typical question on jeopardy?
disambiguated question 2: how long do contestants get to answer a final jeopardy question on jeopardy?
disambiguated question 3: how long do contestants get to answer on jeopardy 's online test?
disambiguated question 4: how long do contestants have to answer during the first two rounds of jeopardy?
clarifying question: For which type of questions: a typical question, a final jeopardy question, jeopardy's online test, or during the first two rounds of jeopard? |
| Example_5 | ambiguous question: Who is the longest serving manager in the premier league?
disambiguated question 1: who is the longest serving manager in the premier league of all time in terms of time?
disambiguated question 2: who is the longest serving manager in the premier league of all time in terms of number of games?
clarifying question: In terms of what: time, or the number of games? |
| Example_6 | ambiguous question: Who sang the original do you love me?
disambiguated question 1: who is the band that sang the original do you love me in 1962?
disambiguated question 2: who is the singer that sang the original do you love me in for the contours in 1962?
disambiguated question 3: who are the characters that sang the original do you love me in the fiddler on the roof?
disambiguated question 4: who are the singers that sang the original do you love me in the 1971 fiddler on the roof film?
clarifying question: Which one: the band in 1962, the singer in the contours in 1962, the characters in the fiddler on the roof, or the singer in the 1971 fiddler on the roof film? |

Table 9: The few-shot examples used for the prompt of the InstructGPT. These examples are concatenated with the instruction in certain order as mentioned in Section 4.

## CQ annotation instruction (Eng)

### Task Definition

Our data will be used to support research in the Natural Language Processing area, specifically for the Open-Domain Question Answering domain.

Our task is to generate **Clarifying Question (CQ)** for the given **Ambiguous Question (AQ)** and **Disambiguated Questions (DQ)**.

- **Ambiguous Question (AQ)** is a question that can have multiple interpretations and multiple plausible answers.

  **Ambiguous Question**: Who is the current chairman of african union commission?

  ○ EX) Above AQ is ambiguous because there can be multiple interpretations of "Current".

- **Disambiguated Questions (DQ)** are the questions that are **edited from the AQ**, and each DQ should have a unique interpretation and unique answer. Orange part of the DQs are the part that is different from the original AQ. Each DQ has the corresponding answer below.

  ☐ Exclude DQ 1

  **Disambiguated Question 1**: who is the 4th chairman of african union commission ?

  **Answer 1**: Moussa Faki / Moussa Faki Mahamat

  ☐ Exclude DQ 2

  **Disambiguated Question 2**: who is the 3rd chairman of african union commission ?

  **Answer 2**: Nkosazana Clarice Dlamini-Zuma / Nkosazana Dlamini-Zuma

  ☐ Exclude DQ 3

  **Disambiguated Question 3**: who is the 2nd chairman of african union commission ?

  **Answer 3**: Jean Ping

  ○ By editing AQ with Orange texts, each DQ became unambiguous so that it has only one interpretation and unique answer.

- **Clarifying Question (CQ)** is a question that "Clarify" AQ, and gives options that lead to corresponding DQs. **Generally, each option is the orange part of each corresponding DQ.**

  ○ CQ consists of the "Category" part and the "options" part

  ○ e.g.) Which chairman: 4th, 3rd, or 2nd?

(a) CQ revision instruction page 1.

### Interface

For each of the given AQ and DQs on the left side of the interface, you will see the "CQ generated by Large Language Model" which is the output of our model. Your task is to **check if the "CQ generated by LLM" has the right categories** and **the right options for corresponding DQs**. If there is an error in "CQ generated by LLM", please revise the CQ after carefully reading the following instructions.

- **"Revise" button**: If you revised the given "CQ generated by LLM", please click revise button after revision.
- **"No problem" button**: If you think the given CQ is well-written and each option in the CQ leads to the corresponding DQs, please click "No Problem" button.
- **"Report" button**: If you think the given AQ has only a single answer, please click "Report" button.

**You should click either "Revise", "No Problem" or "Report" button for each question and then move to the next question.**

**Revise Clarified Question**

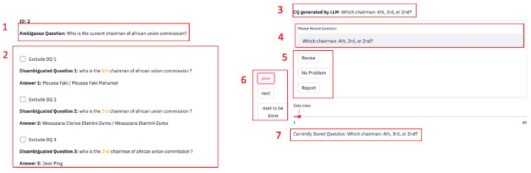

1. Ambiguous Question (AQ)
2. Disambiguated Question (DQ)
3. CQ generated by LLM: The output of our model.
4. Revise CQ: Initialized with "CQ generated by LLM". If you revise the model's output and saved your work, the default output will be changed to the changed one.
5. Submission Buttons:
6. Move buttons: "prev" button will lead you to the previous data. "next" button will lead you to the next data. "next undone" button will lead you to the data that was never revised or checked.
7. This prompt shows the currently saved CQ by you.

(b) CQ revision instruction page 2.

### Instructions

1. Be careful with the **preposition words** before "which" (e.g. "Which year" vs "**As of** which year")

   "As of" case: ID8

   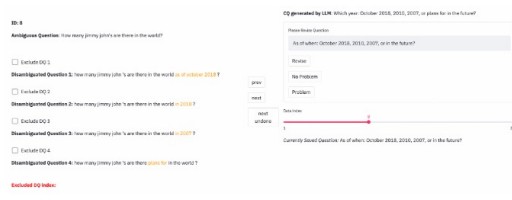

2. Be careful with the category word after "which" (e.g. "Which **season**" vs "Which **year**") **"one" is a good category word to use if it's difficult** determine **the category word** (e.g. "Which one: leader, party, or faction?") "cases where options can not be categorized."

3. The category word should **not be repeated in the options.** (e.g. "Which year: 2010 or 2017?" is good, but "Which year: Year 2010 or year 2017?" is not.)

4. Each option should match a disambiguated question. For multiple disambiguated questions with the **same meaning and** the **same answer**, check "**exclude**" for disambiguated questions to be removed. In the end, **the number of unchecked disambiguated questions and the number of options should be the same.**"

   a. For the case below, DQ 1 and DQ 4 have the same answer. However, since DQ 1 and DQ 4 have different meanings, **you should not exclude** any DQ.

   **Ambiguous Question**: How long do contestants get to answer on jeopardy?

   ☐ Exclude DQ 1

   **Disambiguated Question 1**: how long do contestants get to answer a typical question on jeopardy ?

   **Answer 1**: five seconds

(c) CQ revision instruction page 3.

   ☐ Exclude DQ 2

   **Disambiguated Question 2**: how long do contestants get to answer a final jeopardy question on jeopardy ?

   **Answer 2**: 30 seconds

   ☐ Exclude DQ 3

   **Disambiguated Question 3**: how long do contestants get to answer on jeopardy 's online test ?

   **Answer 3**: 15 seconds

   ☐ Exclude DQ 4

   **Disambiguated Question 4**: how long do contestants have to answer during the first two rounds of jeopardy ! ?

   **Answer 4**: 5 seconds

5. Please **leave the options in the given order** (each option corresponds to each DQ). (e.g. For "Which year: 2012, 2010, or 2017?" 2012 corresponds to DQ1, 2010 corresponds to DQ2, and 2017 corresponds to DQ3.)

(d) CQ revision instruction page 4.

Figure 4: The instructions provided to the recruited annotators for CQ revision.

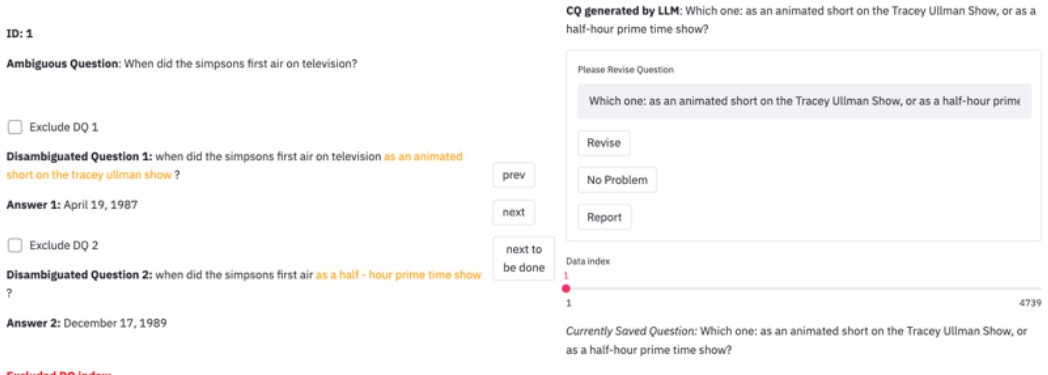

## Revise Clarified Question

**ID: 2095**

CQ generated by LLM: Which takeover: the Russian Red Army, unmarked armed men, with a treaty, or pro-Russian groups?

**Ambiguous Question**: When was the ukraine taken over by russia?

Please Revise Question

> Which takeover: the Russian Red Army, unmarked armed men, with a treaty, or pro-I

☐ Exclude DQ 1

Revise

**Disambiguated Question 1**: when was the ukraine taken over by the russian red army ?

**Answer 1**: 1920

prev

No Problem

next

Report

☐ Exclude DQ 2

next to be done

Data index

**Disambiguated Question 2**: when was the ukraine 's crimean parliamentary building taken over by unmarked armed men ?

2095

1

4739

**Answer 2**: 27 February 2014

☐ Exclude DQ 3

**Disambiguated Question 3**: when was the ukraine 's crimea and sevastopol taken over by russia with a treaty ?

**Answer 3**: 18 March 2014

☐ Exclude DQ 4

**Disambiguated Question 4**: when was the ukraine 's oblasts of donetsk and luhansk taken over by pro - russian groups ?

**Answer 4**: March and April 2014 / Throughout March and April 2014

**Excluded DQ index:**

(a) Interface page for CQ revision example 1.

≡

## Revise Clarified Question

**ID: 1**

CQ generated by LLM: Which one: as an animated short on the Tracey Ullman Show, or as a half-hour prime time show?

**Ambiguous Question**: When did the simpsons first air on television?

Please Revise Question

> Which one: as an animated short on the Tracey Ullman Show, or as a half-hour prime

☐ Exclude DQ 1

Revise

**Disambiguated Question 1**: when did the simpsons first air on television as an animated short on the tracey ullman show ?

prev

No Problem

**Answer 1**: April 19, 1987

next

Report

☐ Exclude DQ 2

next to be done

Data index

**Disambiguated Question 2**: when did the simpsons first air as a half - hour prime time show ?

1

1

4739

**Answer 2**: December 17, 1989

*Currently Saved Question:* Which one: as an animated short on the Tracey Ullman Show, or as a half-hour prime time show?

**Excluded DQ index:**

(b) Interface page for CQ revision example 2.

Figure 5: Interface of qualification and manual editing stage for CQ revision.

| Split | Categories (in the order of frequency) |
|---|---|
| Train | version, year, type, information, time |
| Validation | version, type, time, year, information |
| Test | version, type, information, year, time |

Table 10: Most frequent categories in CAMBIGNQ.

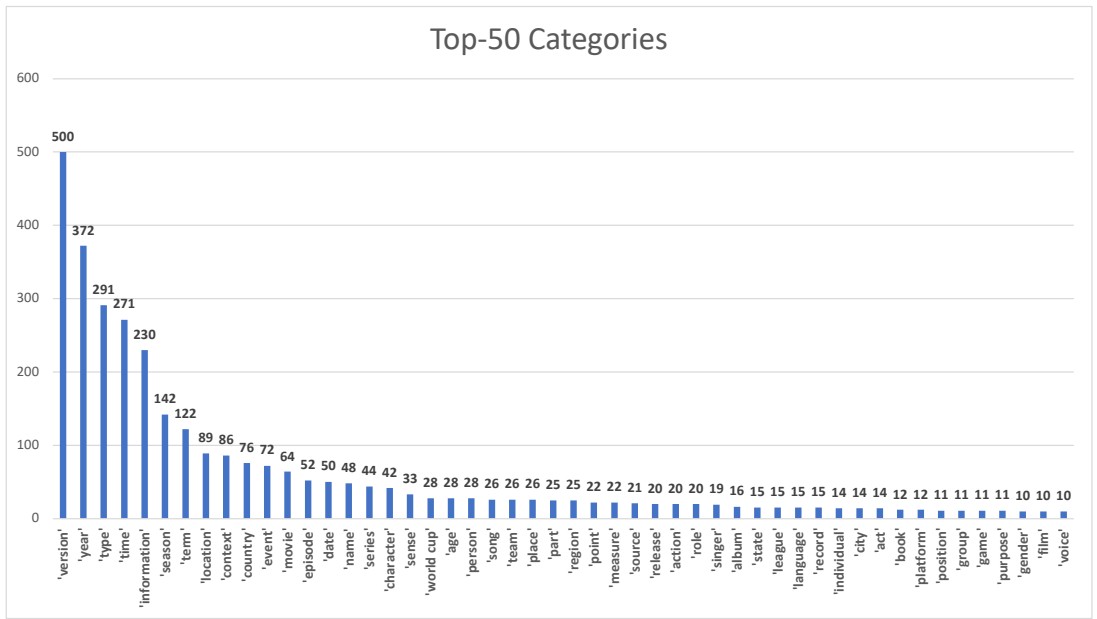

Figure 6: Top-50 categories in CAMBIGNQ.

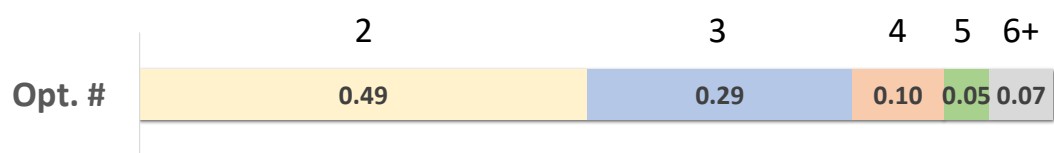

Figure 7: Number of options distribution in CAMBIGNQ.

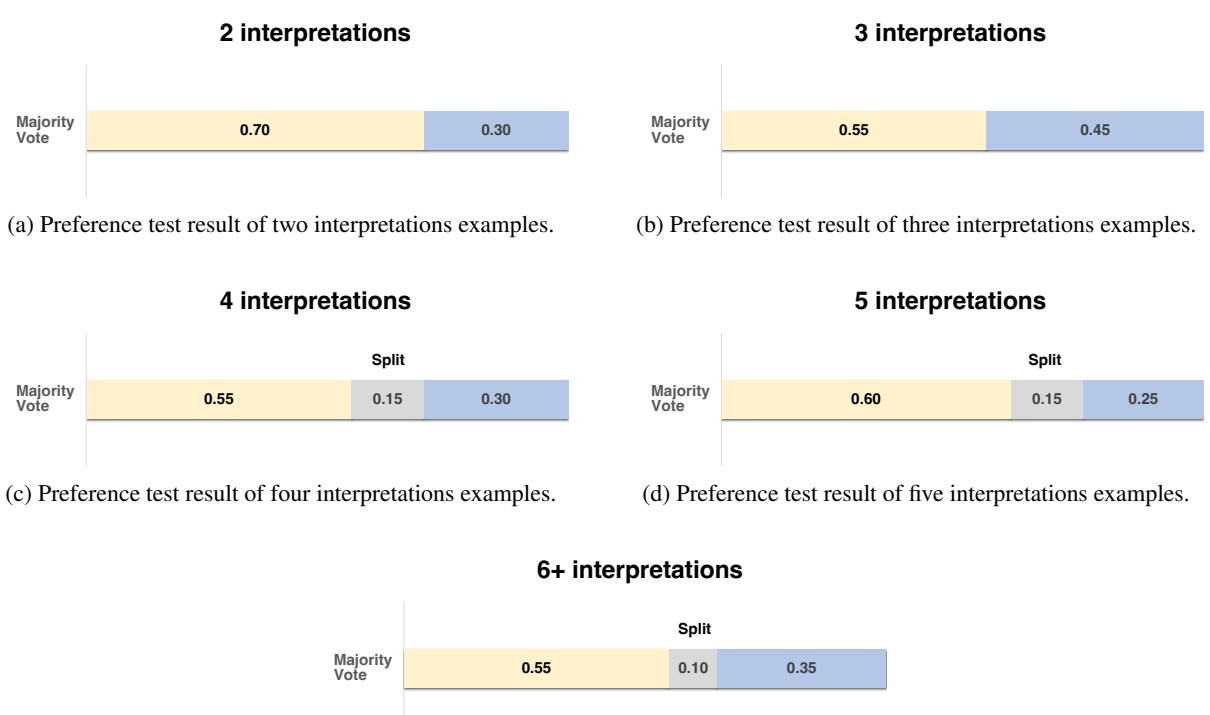

Thank you for participating in this work. Our data will be used to support research in the Natural Language Processing.

Suppose you ask a question on the search engine. However, the question has multiple possible answers.
These answers are shown to you in 2 different ways (Options A and B), and your job is to tell us which option you prefer as a user of the search engine.

Please select equal only when it is really difficult to judge.

**Option A:** (1) You are asked a short clarification question, (2) you select a reponse, and (3) the correct answer is shown to you.

**Option B:** (1) All possible answers are shown to you, one after another.

(a) Instructions given to MTURK workers.

(b) MTURK interface page example.

Figure 8: Interface for preference test for MTURK workers.

**2 interpretations**

Majority Vote — 0.70 | 0.30

(a) Preference test result of two interpretations examples.

**3 interpretations**

Majority Vote — 0.55 | 0.45

(b) Preference test result of three interpretations examples.

**4 interpretations**

Split

Majority Vote — 0.55 | 0.15 | 0.30

(c) Preference test result of four interpretations examples.

**5 interpretations**

Split

Majority Vote — 0.60 | 0.15 | 0.25

(d) Preference test result of five interpretations examples.

**6+ interpretations**

Split

Majority Vote — 0.55 | 0.10 | 0.35

(e) Preference test result of examples with more than five interpretations.

Figure 9: The preference test results for each group with a different number of interpretations.