# OpenReview forum: "Asking Clarification Questions to Handle Ambiguity in Open-Domain QA"
_EMNLP/2023/Conference — EMNLP 2023 Findings_

### Official Review · Reviewer_SP89 · 2023-08-02

**Soundness:** 4

**Excitement:**

3: Ambivalent: It has merits (e.g., it reports state-of-the-art results, the idea is nice), but there are key weaknesses (e.g., it describes incremental work), and it can significantly benefit from another round of revision. However, I won't object to accepting it if my co-reviewers champion it.

**Paper Topic And Main Contributions:**

This paper constructs an open-domain QA dataset (CAMBIGNQ) with clarification questions for resolve the ambiguity. The dataset is built upon an existing dataset (AMBIGNQ). The only difference between these two datasets are that the newly constructed one asks clarification question while the existing one asks disambiguated questions for each possible interpretation. This paper reports the human evaluation on the preference between these two ways for resolving ambiguity in QAs. The results support their arguments that this new setting is more preferable.

In addition to the dataset, this work establishes benchmark performance on three tasks: ambiguity detection, clarification generation and clarification-based question answering. The experimental results reveal the difficulty of these tasks, thus show the need of this new dataset  for future development.

**Questions For The Authors:**

- One key contribution is to generate clarification questions from disambiguated questions. The InstructGPT with few-shot learning shows limited performance. Performance of some intuitive models would be necessary to be reported, such as rule-based approach which extracts new words in disambiguated questions compared to the ambiguous questions and then consider them as "options". Other approaches to be considered is to fine-tune BERT for example. Did you conduct them?

- The clarification question follows a fixed template, i.e. “Which [category]: [option1], [option2], ..., or [optionn]?” In human communication, there could be other ways which may be more natural and fluent given a certain context. Why did you choose this template, and how to compare with other possibilities?

- What do you mean by "reranked related paragraphs" in line 242 and 504? (Reranked by what?)

**Reasons To Accept:**

- Construct a new open-domain QA dataset for resolving ambiguity by asking clarification questions. This dataset can be used for following work to improve the modeling performance.
- Demonstrate the advantage of asking clarification questions than disambiguated questions. This effort encourages future work towards generating clarification question.
- Report solid benchmark performance. These experiments reveal the challenge of these tasks.
- Evaluate InstructGPT's ability on generate clarification questions from disambiguated questions through in-context few-shot learning. The insufficient performance demands better methods.

**Reasons To Reject:**

- Some experimental design is not convincing. For example, in Sec 6.1, they use BERT model for one setting and BART model for the other, so the difference between these two results may not come from whether the predicted answers are useful or not, since it may come from the different capacity of these two models. Another one is that in Table 5, the experiment with ground truth CQ is required, because this will identify whether the low performance in Table 5 is caused by the QA modeling or the generated CQ quality.
- The benchmark experiments are conducted with BERT, BART which don't represent a SOTA performance. These experimental results are not convincing that these tasks are difficult for more SOTA models such as XLNET, GPT-3, GPT-3.5, GPT-4.
- They only consider one fixed template for asking clarification question, which limits the scope of this work.

**Reproducibility:**

3: Could reproduce the results with some difficulty. The settings of parameters are underspecified or subjectively determined; the training/evaluation data are not widely available.

**Reviewer Confidence:**

4: Quite sure. I tried to check the important points carefully. It's unlikely, though conceivable, that I missed something that should affect my ratings.

---

> ### Author Rebuttal · Authors · 2023-08-28
>
> Thanks for acknowledging as strengths the compelling motivation for the proposed task formulation to handle AQs, a systematic presentation of experiment results, and the release of the dataset and the code.
> Following your suggestion, we will provide more analyses of results along with additional experiments:
>
> ### **Response to Reject-1**
> >Some experimental design is not convincing. For example, in Sec 6.1, they use BERT model for one setting and BART model for the other, so the difference between these two results may not come from whether the predicted answers are useful or not, since it may come from the different capacity of these two models. Another one is that in Table 5, the experiment with ground truth CQ is required, because this will identify whether the low performance in Table 5 is caused by the QA modeling or the generated CQ quality.
>
> We appreciate your suggestion for additional analyses and experiments. We first would like to clarify that the choice of different models in Section 6.1 is necessary, given the difference in the approaches to ambiguity detection. BERT (for classification) is used when ambiguity is directly detected through binary classification. BART (for text generation) is used when ambiguity is detected by first generating possible answers and checking whether the number of answers is greater than 1. While we think the former is more natural, the latter is used in related works, so we report the performance as well.
>
> Our initial upper bound case–using CQ generated with ground-truth (GT) answers for AQ–represented a tighter upper bound. We agree with your idea to use GT CQ as a more relaxed upper bound to explain the low performance observed in Table 5. Additional experiments prompting GT CQs with the reader model reveal suboptimal results, which indicates that the low performance in Table 5 is due to the QA modeling. Also, using the SOTA model in the pipeline still shows insufficient performance (See Response to Reject-2 for details) which shows the inherent challenge of our task and further emphasizes the need for qualified datasets and approaches like ours. We’ll include additional experiments in the revision to this effect.
>
> | | CQ finetuned BART | | | |
> |-----------------------------------------------|:-----------------:|:--------:|:--------:|:------:|
> | | Pre. | Rec. | F1 | # Ans. |
> | CQ used to clarify the AQ | | | | |
> | CQ generated with Ground Truth Answers for AQ | 47.5 | 49.5 | 48.5 | 2.5 |
> | Ground Truth CQ | **58.0** | **53.8** | **55.8** | 2.5 |
>
> ### **Response to Reject-2**
> >The benchmark experiments are conducted with BERT, BART which don't represent a SOTA performance. These experimental results are not convincing that these tasks are difficult for more SOTA models such as XLNET, GPT-3, GPT-3.5, GPT-4.
>
> We conducted additional experiments using InstructGPT as the reader model in Section 6.3. We also framed our task as a conversation between the user and the models, using ChatGPT for this purpose.
>
> Despite employing state-of-the-art (SOTA) models, the performance remains suboptimal, which emphasizes the challenging nature of our task. While we acknowledge that not comparing our method with SOTA models may weaken our claims that these tasks are difficult, we want to emphasize our primary focus of our paper:
> 1. Proposing CQs as an alternative to DQs for handling ambiguity in ODQA
> 2. Providing a qualified dataset for future research
> 3. Introducing a framework for generating and evaluating CQs.
>
> Given these objectives, we chose BART-large for its free availability and public checkpoints. Nevertheless, we will discuss the suboptimal performance of SOTA models in the extra page made available upon acceptance to further underscore the challenging nature of our task.
>
> |               |      | CQ finetuned BART |      |          |
> |---------------|:----:|:-----------------:|:----:|:--------:|
> |               | Pre. |        Rec.       |  F1  |    EM    |
> | AQ revised by |      |                   |      |          |
> | Ground Truth CQ         | 58.0 |      **53.8**     | 55.8 | **35.8** |
>
> |                    ||                InstructGPT             |||
> |---------------|:----:|:-----------:|:----:|:--------:|
> |               | Pre. |     Rec.    | F1   |    EM    |
> | AQ revised by |      |             |      |          |
> | Ground Truth CQ         |  7.4 |   **60.0**  | 13.1 | **43.2** |
>
>
> |            | ChatGPT |          |      |          |
> |------------|:-------:|:--------:|:----:|:--------:|
> |Ground Truth CQ|   Pre.  |   Rec.   |  F1  |    EM    |
> | zero-shot  |   8.0   | **64.5** | 14.3 | **50.8** |
> | four-shot  |   11.3  | **64.0** | 19.2 | **49.9** |
>
> Please note that the low precision in the SOTA model’s results can be attributed to the models generating answers at the sentence level. Specifically, InstructGPT averaged 27.3 words, ChatGPT in a zero-shot setting averaged 25.4 words, and ChatGPT in a four-shot setting averaged 19.9 words, compared to the gold answers averaging 2.6 words. We also report the Exact Match metric, which checks whether the gold answer is included in the generated sentences.
>
> ### **Response to Reject-3**
> > They only consider one fixed template for asking clarification question, which limits the scope of this work.
>
> We have considered several CQ formats and decided to use the current format as the only format, as it is the simplest format containing all absolutely necessary information (options) that we could think of. We believe this format not only makes the problem easier but also minimizes the amount of text that users need to read. And since it’s the simplest format containing all necessary information, it is actually more general and applicable than a more detailed and tailored format.
>
> ### **Answer to Question-1**
> >One key contribution is to generate clarification questions from disambiguated questions. The InstructGPT with few-shot learning shows limited performance. Performance of some intuitive models would be necessary to be reported, such as rule-based approach which extracts new words in disambiguated questions compared to the ambiguous questions and then consider them as "options". Other approaches to be considered is to fine-tune BERT for example. Did you conduct them?
>
>
> Our objective with respect to the dataset construction was to somehow construct a quality dataset of CQs, rather than to find and propose an optimal way to do so. Thus, we do not think comparing with other means to construct the datasets is high on the priority list given the limited space. But, we did try several approaches during a pilot study including a rule-based approach, which turned out to be quite noisy. We found the current approach of using InstructGPT with manual editing to be sufficient for constructing the dataset we aimed for.
>
>
> ### **Answer to Question-2**
> >The clarification question follows a fixed template, i.e. “Which [category]: [option1], [option2], ..., or [optionn]?” In human communication, there could be other ways which may be more natural and fluent given a certain context. Why did you choose this template, and how to compare with other possibilities?
>
> As stated in our response to Response to Reject-3, we expect the simplicity of our CQ format to enhance user experience while making the construction process easier. While other formats might offer fluency or naturalness, they could add complexity to the construction process without increasing informativeness, as this is this simple format contains all absolutely necessary information (options).
>
>
> ### **Answer to Question-3**
> >What do you mean by "reranked related paragraphs" in line 242 and 504? (Reranked by what?)
>
> Relevant passages are retrieved based on the ambiguous questions. For both L242 and L542 cases, AQs are revised by CQs and subsequently disambiguated. Passages are then reranked based on the revised AQs. We used the publicly available sentence cross-encoder model 'MiniLM,' pretrained on the MSMARCO dataset, for this task. This information will be included in the revised manuscript.

---

### Official Review · Reviewer_1HwZ · 2023-08-02

**Typos Grammar Style And Presentation Improvements:** n/a
**Soundness:** 3

**Excitement:**

2: Mediocre: This paper makes marginal contributions (vs non-contemporaneous work), so I would rather not see it in the conference.

**Missing References:**

n/a

**Paper Topic And Main Contributions:**

The paper proposes an innovative approach to handle ambiguous questions (AQ) in open-domain QA by introducing clarifying ambiguous natural questions (CAmbigNQ), a dataset consisting of 5,653 AQs with relevant passages, possible answers, and clarification questions (CQ). Unlike previous methods that resolved ambiguities by **directly** generating disambiguated questions (DQs), the authors present a novel method that first ask CQ before generating an answer. The pipeline consists of three tasks: ambiguity detection, clarification question generation, and clarification-based QA. Experimental results are presented, emphasizing the need for further enhancements.

**Questions For The Authors:**

1. While CQ improves user preference (as shown in experiment 1), 33% of people preferred not to generate a CQ. Could this indicate the generated CQs could be misleading or incorrect? Why do human annotators do not prefer those CQs? It would be beneficial to understand the underlying reasons for this preference.

And at the very least, generating CQs should ideally (at least) not alter human preference, as more information is provided compared to only AQ and DQ provided. However, the fact that human preference decreases by 33% when generating CQs raises questions about the necessity and effectiveness of producing these CQs.

2. The paper relies on large language models such as ChatGPT to generate CQs. I am curious about the ability of a model like ChatGPT to **directly** solve AQ, instead of using generated CQs to train small language model? Can ChatGPT solve AQ in a form similar to chain-of-thought, for example, a chain consisting of (1) ask CQ (2) generate DQ (3) answer the AQ. To summarize, what is the ChatGPT performance on such task? Could ChatGPT perform better than this complex pipeline?

3. line 394 stated *“since predicted answers for AQ have been shown to be helpful for DQ based approaches”*, however, as shown in Table
3 and Table 6, it seems *No Answers for AQ* makes better performance than *Predicted Answers for AQ* on the ambiguity detection and end-QA task. Does it mean your observation contradict to the observation from existing literature? Can you explain this discrepancy? Or please correct me if I misunderstand this.

4. Since *category* name are mostly short, so, does the use of BLEU or EM score as evaluation metrics reflect the correctness of generated CQs accurately? How does it align with human evaluations? And also same question as the evaluation of CQs, did authors perform human evaluation or semantic-based evaluation on CQs, since the CQs is much longer than *category*?

5. As shown in Table 5, it seems adding this CQ generation, as an intermediate process, does not help much on end-to-end QA performance, i.e., predicting answers for AQ. So, why do authors think generating CQ is necessary if people still mostly care about the end-to-end QA performance?

6. The proposed method makes good ablation study on using or not using CQ in the question answering generation process. However, the paper does not compare with SoTA performance on AmbigQA.

Leaderboard: https://nlp.cs.washington.edu/ambigqa/leaderboard.html


Minor:

7. For CQ generation evaluation, how many references are provided? Do authors construct multi-reference for evaluation to improve the robustness?

8. Do authors try different prompts (as shown in line 239 and line 289) to generate CQ using ChatGPT?

**Reasons To Accept:**

**-- Novel Approach:** The authors introduce a new method by shifting from conventional disambiguated question generation to a clarification questions-based method, aligning more closely with real-world applications.

**-- New Dataset:** The creation of the CAmbigNQ dataset to support research in ambiguity in QA represents a valuable asset for the community.

**-- Writing Quality:** The paper is clear, well-written, and effectively uses examples to explain the concepts, enhancing readability and understanding.


**Reasons To Reject:**

**-- Doubtful Motivation:** While CQ improves user preference (as shown in Experiment 1), 33% of people preferred not to generate a CQ, which raises questions about the necessity and effectiveness of producing these CQs.

**-- Insufficient Comparison with SoTA:** The paper lacks comparison with state-of-the-art methods, which is essential to gauge the true effectiveness of the proposed approach.

**-- Questions about Methodology:** Several questions arise concerning the methodology, including the choice of evaluation metrics, the contradiction in observations, and the real necessity of CQ generation for end-to-end performance. These need to be clearly addressed to strengthen the paper (see questions for more details).


**Reproducibility:**

4: Could mostly reproduce the results, but there may be some variation because of sample variance or minor variations in their interpretation of the protocol or method.

**Reviewer Confidence:**

4: Quite sure. I tried to check the important points carefully. It's unlikely, though conceivable, that I missed something that should affect my ratings.

---

> ### Author Rebuttal · Authors · 2023-08-28
>
> Thanks for acknowledging as strengths the novel approach to handling AQs under realistic scenarios, our dataset, and the clarity of our writing.
>
> ### **Response to Reject-1**
> >**Doubtful Motivation**: While CQ improves user preference (as shown in Experiment 1), 33% of people preferred not to generate a CQ, which raises questions about the necessity and effectiveness of producing these CQs.
>
> Please see our **Answer to Question-1**.
>
>
> ### **Response to Reject-2**
> >**Insufficient Comparison with SoTA**: The paper lacks comparison with state-of-the-art methods, which is essential to gauge the true effectiveness of the proposed approach.
>
> As per your suggestion, we have conducted additional experiments using InstructGPT—a model trained for NLP tasks rather than conversation like ChatGPT—as a reader in place of BART in Section 6.3. In this experiment, we employed ground-truth CQs instead of model-generated CQs to measure the best possible performance in the ideal setting, as Reviewer SP89 suggested.
> The two tables below indicate that the InstructGPT showed improved performance compared to our baseline (for which we also report CQ fine-tuned BART results in the same setup). However, there remains substantial room for improvement, underscoring the challenging nature of our task. Please note that the low precision in InstructGPT’s results can be attributed to the model generating answers at the sentence level, which averages 27.3 words, compared to the gold answers averaging 2.6 words. We also report the Exact Match metric, which checks whether the gold answer is included in the generated sentences.
>
>
> Initially, we chose BART-large for its free availability and public checkpoints, making it a suitable starting point for future research.
> Furthermore, direct comparisons with SOTA methods in AmbigQA are not feasible due to differences in tasks (Please see answer to Questions-6 for details).
> |               | |      InstructGPT    |      |          |
> |---------------|:-----------:|:--------:|:----:|:--------:|
> |               |     Pre.    |   Rec.   | F1   |    EM    |
> | AQ revised by |             |          |      |          |
> | Ground Truth CQ         |     7.4     | **60.0** | 13.1 | **43.2** |
>
> |               |      | **CQ finetuned BART** |      |      |
> |---------------|:----:|:---------------------:|:----:|:----:|
> |               | Pre. |          Rec.         |  F1  |  EM  |
> | AQ revised by |      |                       |      |      |
> | Ground Truth CQ         | 58.0 |          **53.8**         | 55.8 | **35.8** |
>
>
> ### **Response to Reject-3**
> >**Questions about Methodology**: Several questions arise concerning the methodology, including the choice of evaluation metrics, the contradiction in observations, and the real necessity of CQ generation for end-to-end performance. These need to be clearly addressed to strengthen the paper (see questions for more details).
>
> Thank you for your comprehensive feedback. We will address questions individually:
> - The choice of evaluation metrics (Please see **Answer to Question-4**)
> - The contradiction in observation (Please see **Answer to Question-3**)
> - The real necessity of CQ generation for end-to-end performance (Please see **Answer to Question-5**)
>
>
> ### **Answer to Question-1**
> >1. While CQ improves user preference (as shown in experiment 1), 33% of people preferred not to generate a CQ. Could this indicate the generated CQs could be misleading or incorrect? Why do human annotators do not prefer those CQs? It would be beneficial to understand the underlying reasons for this preference.
> And at the very least, generating CQs should ideally (at least) not alter human preference, as more information is provided compared to only AQ and DQ provided. However, the fact that human preference decreases by 33% when generating CQs raises questions about the necessity and effectiveness of producing these CQs.
>
> Just to clarify, the alternatives being compared are: **“1 AQ + n DQs + n answers for the DQs” vs “1 AQ + 1 CQ + 1 answer based on the user response to the CQ”.** **No additional information is provided in CQ cases compared to DQ cases**. We will make this more clear in the paper.
>
> Yes, 33% of people indeed preferred DQs over CQs. However, as shown in Fig 3, 59% of people preferred CQs over DQs. As discussed in Sec 5, people generally preferred CQ when there are many possible interpretations, but when the options are few, some people preferred DQs. This is because the benefits of using less screen space and reducing the amount of text to be read by people are magnified as the number of DQs increases.
>
>
> ### **Answer to Question-2**
> >2. The paper relies on large language models such as ChatGPT to generate CQs. I am curious about the ability of a model like ChatGPT to directly solve AQ, instead of using generated CQs to train small language model? Can ChatGPT solve AQ in a form similar to chain-of-thought, for example, a chain consisting of (1) ask CQ (2) generate DQ (3) answer the AQ. To summarize, what is the ChatGPT performance on such task? Could ChatGPT perform better than this complex pipeline?
>
> Firstly, our problem setting consists of 1) a user asking an ambiguous question, 2) the model responding with a clarification question, 3) the user selecting an option, and 4) the model generating a distinct answer based on the selected option. **This is not a sequential reasoning process but an interactive one based on user responses, making it unsuitable for chain-of-thoughts.**
> We conducted additional experiments with ChatGPT per your comment, by prompting our task as a conversation between the user and models. We found that ChatGPT struggled to generate distinct answers in both zero-shot and few-shot scenarios which again showcases the challenging nature of our tasks and the need for further research. Similar to experiments conducted by using InstructGPT, ChatGPT yields low precision due to the sentence level generation, averaging 25.4 words in the zero-shot and 19.9 words in the four-shot scenario.
>
> |            | **ChatGPT** |          |      |          |
> |------------|:-----------:|:--------:|:----:|:--------:|
> | Ground Truth CQ |     Pre.     |    Rec.   |  F1  |    EM    |
> | zero-shot  |     8.0     | **64.5** | 14.3 | **50.8** |
> | four-shot  |     11.3    | **64.0** | 19.2 | **49.9** |
>
> ### **Answer to Question-3**
> > 3. line 394 stated “since predicted answers for AQ have been shown to be helpful for DQ based approaches”, however, as shown in Table 3 and Table 6, it seems No Answers for AQ makes better performance than Predicted Answers for AQ on the ambiguity detection and end-QA task. Does it mean your observation contradict to the observation from existing literature? Can you explain this discrepancy? Or please correct me if I misunderstand this.
>
> We did mean that predicted answers for AQ were helpful for DQ-based approach. But this paper proposes a CQ-based approach, which is a different task setup to deal with AQ. That is, unlike DQ-based approaches in which all possible DQs and respective answers are found for the given AQ, CQ-based approach involves generating a CQ and finding 1 answer to AQ based on the user response to the CQ. We believe terms like “CQ-based approach” and “DQ-based approach” are causing confusion. We will clarify this in the paper.
>
> ### **Answer to Question-4**
> > 4. Since category name are mostly short, so, does the use of BLEU or EM score as evaluation metrics reflect the correctness of generated CQs accurately? How does it align with human evaluations? And also same question as the evaluation of CQs, did authors perform human evaluation or semantic-based evaluation on CQs, since the CQs is much longer than category?
>
> In light of supporting future work on the new task we propose, we chose to design automatic evaluation metrics so that other researchers can easily and fairly compare against our baseline results. Thanks for your suggestion to use a semantic-based evaluation metric, which still aligns with our goal and provides an additional means of evaluation. As per your suggestion, we measured BERTSCORE (Zhang et al., 2020), and these results and additional analysis will be included in the revised manuscript. The error analysis in Section 6.2 also gives insight to the future researcher that automatic evaluation might capture the relative quality of generated CQ but not implicit quality.
> | Input in addition to AQ and RPs | CQ | | Category | | Options | | | |
> |---------------------------------|:------:|:---------:|:--------:|:------:|:-------:|:----:|:----:|:------:|
> | | BLEU-4 | BERTSCORE | EM | BLEU-1 | Pre. | Rec. | F1 | Avg. # |
> | No answers for AQ | **7.9** | **88.9** | 20.2 | 47.3 | 37.4 | 18.2 | 24.5 | 2.0 |
> | Predicted answers for AQ | **7.9** | **88.9** | 22.8 | 44.0 | 36.9 | 19.0 | 25.1 | 2.0 |
> | Ground Truth answers for AQ | **15.4** | **89.6** | 25.2 | 46.9 | 34.3 | 34.4 | 34.3 | 3.7 |
>
>
> ### **Answer to Question-5**
> > 5. As shown in Table 5, it seems adding this CQ generation, as an intermediate process, does not help much on end-to-end QA performance, i.e., predicting answers for AQ. So, why do authors think generating CQ is necessary if people still mostly care about the end-to-end QA performance?
>
> As previously discussed, the preference test shows that CQ can improve user experience, with 59% of people preferring CQ over DQs. Accordingly, the room for improvement in the end-to-end performance can be seen as a reason to conduct further research in this direction rather than discarding this setup.
>
> ### **Answer to Question-6**
> > 6. The proposed method makes good ablation study on using or not using CQ in the question answering generation process. However, the paper does not compare with SoTA performance on AmbigQA.
>
> We cannot compare against the SOTA performance on AmbigQA, because the tasks are different— although both tasks were proposed to solve the same issue of dealing with AQs. In its essence, AmbigQA (DQ-based approach to deal with AQ) takes in an AQ and relevant passages and 1) first predicts plausible answers, 2) then generates DQs (to be displayed to the user with each answer). On the other hand, we (CQ-based approach to deal with AQ) take in an AQ with relevant passages and 1) first generate CQ, 2) then generate an answer for the AQ based on the response to CQ, where the CQ and the response to CQ ideally reveal the single correct interpretation of the AQ.
>
> ### **Answer to Question-7**
> > 7. For CQ generation evaluation, how many references are provided? Do authors construct multi-reference for evaluation to improve the robustness?
>
> We chose a simple CQ format to enhance user experience and ease the construction process. Although more complex formats are possible, our current format was found to be effective.
>
> ### **Answer to Question-8**
> > 8. Do authors try different prompts (as shown in line 239 and line 289) to generate CQ using ChatGPT?
>
> We experimented with various prompts in a pilot study and selected the ones that performed best to be used in the main experiments.

---

### Official Review · Reviewer_Sdsh · 2023-08-04

**Typos Grammar Style And Presentation Improvements:** I could not see any typos. The paper …
**Soundness:** 4

**Excitement:**

4: Strong: This paper deepens the understanding of some phenomenon or lowers the barriers to an existing research direction.

**Paper Topic And Main Contributions:**

When confronted with an ambiguous question, rather than have the alternatives spelled out as a list of separate
questions, it is more natural to produce a clarification question with concisely expressed options. In this paper the
authors describe the addition to an existing dataset of ambiguous questions, relevant passages, and possible answers, a
clarification question of the preferred type. This was done using InstructGPT with some few-shot examples to generate
possible candidates which were then manually checked and revised.

They distinguish two subtasks relevant to the creation of this dataset: ambiguity detection and clarification question
generation, and one which uses the results of these subtasks, namely finding answers to the clarification question.

Annotators were asked to state a preference between a number of the original verbose individual questions and the more
compact alternative version. This latter was usually preferred, and was invariably preferred when more than three interpretations were possible.

The paper presents preliminary results for ambiguity detection: under one condition, a BERT-based classifier is trained
to distinguish ambiguous from non-ambiguous questions (trained and tested on the original dataset) and under another a
BART-based model is used to predict answers, with the input classified as ambiguous if more than one is predicted. The
first classifier achieved 61.3 F1, the second 34.3.

There are also preliminary results for clarification question generation. Various BART models are trained to predict the
clarification question given (1) the ambiguous one and relevant passages, or (2) the ambiguous question, answers as
predicted by one of the models just described as well as the relevant passages. For comparison, in one condition (3) the
predicted answers were replaced by the actual answers, as a kind of best case. Naturally, the latter case produced the
best results, with no significant variation between 1 and 2.

The final set of experiments tests QA performance on clarification questions, testing four conditions: ambiguity
detection with and without predicted answers and clarification questions created with or without the predicted
answers. Perhaps surprisingly the best results were obtained with no answers for either stage.

The main contributions of the paper are (1) a clear description of the problem (2) provision (public?) of a dataset and (3) initial results for the accuracy of automation of some of the stages, and utility in a QA setting.


**Reasons To Accept:**

This is an interesting set of problems, which the paper presents one type of solution to.

**Reasons To Reject:**

I can't see any reason to reject this paper.

**Reproducibility:**

2: Would be hard pressed to reproduce the results. The contribution depends on data that are simply not available outside the author's institution or consortium; not enough details are provided.

**Reviewer Confidence:**

3: Pretty sure, but there's a chance I missed something. Although I have a good feel for this area in general, I did not carefully check the paper's details, e.g., the math, experimental design, or novelty.

---

> ### Author Rebuttal · Authors · 2023-08-28
>
> Thanks for acknowledging as strengths the clarity of our description, the dataset, and the initial performance of baselines of our proposed task.
>
> We appreciate the comment that you do not see a reason to reject our work. To address the concerns about the reproducibility of our work, we will make our dataset and models publicly available after acceptance.

---

### Meta-Review · Area_Chair_Guum · 2023-09-12

**Recommendation:** 4

**Metareview:**

The paper describes a novel dataset for resolving question ambiguity in open-domain QA and proposes an innovative approach: instead of directly generating disambiguated questions (DQs) for every possible interpretation, Clarification Questions (CQs) are generated to be asked to the user before generating the correct answer.  The dataset builds upon the existing AMBIGNQ and adds appropriate clarification questions for each prompt. CQs are generated automatically, then evaluated and revised by human annotators. Human evaluation of the preference between the two settings (i.e. DQ vs CQ) supports the argument that CQ is preferable. Experiments with BART, however, do not report better performance, which indicates the tasks are challenging. The reviewers initially had different positions but ended up in coming closer on soundness after the author rebuttals.

**Pros.**

The paper presents a novel approach to a still challenging task in QA;

It contributes a novel dataset, useful for future developments in dealing with AQ in QA;

It establishes a benchmark performance on three tasks: ambiguity detection, clarification generation and clarification-based question answering, for easier and better comparison in future works;

It demonstrates the need for better models/ methos to handle ambiguity in QA;

It is fairly well written, it states clearly its objectives, and authors seem to have covered the related literature well.

**Cons:**

The paper lacks an explanation for the use of different models in some experiments (see reviewer SP89 comments). Authors, however, provided satisfactory details in their rebuttal.

experiments with ground-truth CQs are missing, which would better complement and explain the result in Figure 5.  As, authors already run such experiments, they can easily be added to the paper;

The benchmark experiments do not include SOTA models, which makes the claim that the targeted tasks are still a challenge less convincing. For their rebuttal authors performed experiments with InstructGPT and ChatGPT that confirm the claim. This should be included in the paper.




It is recommended that the authors integrate the promised additional materials into the final paper.

---

### Decision · Program_Chairs · 2023-10-07

**Decision:**

Accept-Findings

**Comment:**

The paper describes a novel dataset for resolving question ambiguity in open-domain QA and proposes an innovative approach: instead of directly generating disambiguated questions (DQs) for every possible interpretation, Clarification Questions (CQs) are generated to be asked to the user before generating the correct answer.  The dataset builds upon the existing AMBIGNQ and adds appropriate clarification questions for each prompt. CQs are generated automatically, then evaluated and revised by human annotators. Human evaluation of the preference between the two settings (i.e. DQ vs CQ) supports the argument that CQ is preferable. Experiments with BART, however, do not report better performance, which indicates the tasks are challenging. The reviewers initially had different positions but ended up in coming closer on soundness after the author rebuttals.

**Pros.**

The paper presents a novel approach to a still challenging task in QA;

It contributes a novel dataset, useful for future developments in dealing with AQ in QA;

It establishes a benchmark performance on three tasks: ambiguity detection, clarification generation and clarification-based question answering, for easier and better comparison in future works;

It demonstrates the need for better models/ methos to handle ambiguity in QA;

It is fairly well written, it states clearly its objectives, and authors seem to have covered the related literature well.

**Cons:**

The paper lacks an explanation for the use of different models in some experiments (see reviewer SP89 comments). Authors, however, provided satisfactory details in their rebuttal.

experiments with ground-truth CQs are missing, which would better complement and explain the result in Figure 5.  As, authors already run such experiments, they can easily be added to the paper;

The benchmark experiments do not include SOTA models, which makes the claim that the targeted tasks are still a challenge less convincing. For their rebuttal authors performed experiments with InstructGPT and ChatGPT that confirm the claim. This should be included in the paper.




It is recommended that the authors integrate the promised additional materials into the final paper.